# Pharmacokinetics and Pharmacodynamics of Cefepime in Adults with Hematological Malignancies and Febrile Neutropenia after Chemotherapy

**DOI:** 10.3390/antibiotics10050504

**Published:** 2021-04-29

**Authors:** José C. Álvarez, Sonia I. Cuervo, Edelberto Silva, Jorge A. Díaz, Lorena L. Jiménez, Daniel S. Parra, Julio C. Gómez, Ricardo Sánchez, Jorge A. Cortés

**Affiliations:** 1Facultad de Medicina, Universidad Nacional de Colombia, 111321 Bogotá, Colombia; jcalvarezr@unal.edu.co (J.C.Á); rsanchezpe@unal.edu.co (R.S.); jacortesl@unal.edu.co (J.A.C.); 2Grupo en Enfermedades Infecciosas en Cáncer y Alteraciones Hematológicas (GREICAH), Universidad Nacional de Colombia, 111321 Bogotá, Colombia; esilvag@unal.edu.co (E.S.); jadiazr@unal.edu.co (J.A.D.); lljimenezb@unal.edu.co (L.L.J.); jcgomezrincon@gmail.com (J.C.G.); 3Instituto Nacional de Cancerología (INC)—Empresa Social del Estado, 111511 Bogotá, Colombia; 4Departamento de Farmacia, Facultad de Ciencias, Universidad Nacional de Colombia, 111321 Bogotá, Colombia; dsparrag@unal.edu.co; 5Grupo de Investigación en Enfermedades Infecciosas, Facultad de Medicina, Universidad Nacional de Colombia, 111321 Bogotá, Colombia

**Keywords:** cefepime, cephalosporins, pharmacokinetics, chemotherapy-induced febrile neutropenia, hematologic neoplasms

## Abstract

Patients with chemotherapy-induced febrile neutropenia (CIFN) may have changes in the pharmacokinetics (PK) compared to patients without malignancies or neutropenia. Those changes in antibiotic PK could lead to negative outcomes for patients if the therapy is not adequately adjusted to this. In this, open-label, non-randomized, prospective, observational, and descriptive study, a PK model of cefepime was developed for patients with hematological neoplasms and post-chemotherapy febrile neutropenia. This study was conducted at a cancer referral center, and study participants were receiving 2 g IV doses of cefepime every 8 h as 30-min infusions. Cefepime PK was well described by a two compartment model with a clearance dependent on a serum creatinine level. Using Monte Carlo simulations, it was shown that continuous infusions of 6g q24h could have a good achievement of PK/PD targets for MIC levels below the resistance cut-off point of Enterobacteriaceae. According to the simulations, it is unnecessary to increase the daily dose of cefepime (above 6 g daily) to increase the probability of target attainment (PTA). Cumulative fraction of response (CFR) using interment dosing was suboptimal for empirical therapy regimens against *K. pneumoniae* and *P. aeruginosa*, and continuous infusions could be used in this setting to maximize exposure. Patients with high serum creatinine levels were more likely to achieve predefined PK/PD targets than patients with low levels.

## 1. Introduction

Patients with chemotherapy-induced febrile neutropenia (CIFN) have a higher risk of infection according to the severity and duration of neutropenia [1]. Additionally, they present pathophysiological changes that alter the pharmacokinetic (PK) parameters of hydrophilic antibiotics such as β-lactams; for example, an increase in the volume of distribution ($V_{D}$) and its clearance (CL) that are explained by the presence of a third space, hypoalbuminemia, and cachexia present in these patients, which can, in turn, result in inadequate serum and tissue concentrations of antimicrobials, and eventual therapeutic failures [2].

Knowledge of the variability in PK parameters is useful for antibiotic stewardship programs. Such knowledge serves to optimize the prescription of antibiotics such as β-lactams and vancomycin often required for the management of this particular group of patients, which in turn has an impact on favorable clinical outcomes and cost-effective decisions [3]. Knowledge of the PK parameters of β-lactam antibiotics in patients with CIFN has expanded in recent years [4,5,6] and the variability that such parameters present in this special group have been described concerning other groups of critically ill patients, such as those who receive cefepime in burn units or the ICU [6].

Cefepime (FEP) is a cephalosporin with a broad spectrum of activity [7] and good penetration into various tissues [8]. The microbiological effect of β-lactam antibiotics is associated with the time that the concentration of free drug remains above the MIC ($fT_{>\mathrm{MIC}}$) [9], and this is considered as the PK/PD indicator (PDI) of cefepime.

For patients with CIFN, it is not entirely clear the PDI target associated with the clinical cure, microbiological eradication, or survival [10]. In a study of patients with bacteremia and sepsis [11], target attainment of >80%$fT_{>\mathrm{MIC}}$ with cefepime was associated with a lower risk of adverse outcomes related to bacteriological eradication and clinical cure. In another study of patients with *P. aeruginosa* infections [12], microbiological eradication was found to be associated with achieving a target of $60\%fT_{>\mathrm{MIC}}$ with cefepime. Finally, in another study of patients with Gram-negative bacteremia [13], in-hospital survival was shown to be related to achieving a target between 68 to $74\%fT_{>\mathrm{MIC}}$.

In patients with CIFN, cefepime is recommended as a first-line β-lactam in empirical treatment [14,15]. However, its use has been associated with an increase in mortality with clearly not established causes [16]. The aims of this study were: (i) describe the pharmacokinetic variability of cefepime in these patients and determine the influence of covariates with a population pharmacokinetic model, and (ii) to assess the probability of target attainment (PTA) of various FEP dosing regimens using simulation, to propose optimal dosage regimens for FEP.

## 2. Materials and Methods

### 2.1. Study Design and Ethics

An open-label, non-randomized, prospective, observational, descriptive study was conducted at Instituto Nacional de Cancerología (INC) of Bogota, Colombia, a 150-bed referral institution. The ethics committees of INC and the Faculty of Sciences of the Universidad Nacional del Colombia approved the protocol. Informed consent was obtained from patients and/or their representatives, for all participants.

The inclusion criteria were adults older than 18 years of age with a diagnosis of confirmed de novo hematologic neoplasms, receiving induction or maintenance chemotherapy, treated with cefepime at a dose of 2 g IV every 8 h. Febrile neutropenia was defined as a disorder characterized by: (i) an absolute neutrophil count (ANC) <1000/μL, and (ii) a single oral temperature ≥ 38.3 ${}^{\circ }$C or a temperature ≥ 38.0 ${}^{\circ }$C sustained over 1 h [17].

The exclusion criteria included: (i) pregnancy, (ii) acute kidney injury [18], (iii) chronic kidney disease defined as an estimated glomerular filtration rate (eGFR) <60 $\mathrm{mL}/\min/1.73\;m^{2}$ [19], (iv) liver failure defined as Child-Pugh B or C [20], (v) significant comorbidities in addition to its underlying neoplasm, (vi) heart failure [21], (vii) hypothyroidism, (viii) diabetes mellitus, (ix) being on antimicrobial therapy combined with glycopeptides or aminoglycosides, and (x) polymicrobial bloodstream infection. eGFR was calculated using the CKD-EPI equation [22].

### 2.2. Medication Dosage and Administration

These patients were prescribed cefepime (brand Vitalis^®^) at a dose of 2 g every 8 h, administered in a 30-min infusion. The drug was reconstituted and diluted with 0.9% saline. The infusions were made using an infusion pump.

### 2.3. Blood Sampling

In addition, 5 mL of venous blood was taken using a 22- or 24-gauge catheter, at least 24 h after the start of therapy. Six blood samples were collected per patient at 5 min, 30 min, 90 min, 3 and 6 h after the end of the infusion, and 10 min before starting the next infusion of cefepime. The samples were centrifuged at 3500 rpm for 10 min, and were frozen at −70 °C in cryovials, until the moment of analysis.

### 2.4. Drug Measurement

The determination of plasma concentration of cefepime was carried out by means of microbiological assay [23,24]. The diameters of the growth inhibition halos of *Bacillus subtillis* strain ATCC 6633 were measured. Calibration curves were performed on antibiotic-free serum from cancer patients to optimize test conditions for cefepime [25]. A linear relationship was obtained between inhibition halos and the logarithm of FEP concentration in the range studied (3.125–50 mg/L). The intra- and inter-day precision of the assay, expressed as coefficient of variation (CV), was less than 6.7% and 9.5%, respectively, and the relative bias (to assess accuracy) was less than 9% at all concentration levels evaluated in the validation. The lower limit of quantification (LLOQ) was defined as the lowest concentration with a CV < 20% and was found to be 3.125 mg/mL.

### 2.5. Pharmacokinetic Modeling

A population pharmacokinetic model of cefepime was developed using the Stochastic Approach of the Expectation–Maximization algorithm (SAEM) with the Monolix 2019-R2 software (Lixoft SAS, Antony, France) [26]. We explored one-, two-, and three-compartment structural models with linear elimination, as well as multiple error models (additive, proportional, and combined).

The parameters of each individual were estimated with an exponential equation $P_{i}=P_{\mathrm{pop}}\cdot \exp\left(\eta_{i}\right)$, where $P_{i}$ represents the value of the parameter in the *i*-th individual, $P_{\mathrm{pop}}$ is the typical value of the parameter P in the population, $\eta_{i}$ is the deviation of $P_{i}$ from $P_{\mathrm{pop}}$ with a distribution (0, $\omega^{2}$). Some points were below the LLOQ, and, for this reason, the M4 method for handling censored data described by Beal [27] was used. The likelihood ratio test (LRT) and the corrected Bayesian information criterion (BICc) [28] were used to test different hypotheses related to the selection of the structural, covariate, and error model.

Potential covariates were sex, age, weight, height, body mass index (BMI), body surface area, serum creatinine ($S_{\mathrm{CR}}$), eGFR, serum protein concentration, serum albumin, absolute neutrophil count, and absolute leukocyte count. Various forms of equations with linear, potential, and exponential forms were evaluated for continuous covariates, while an additive equation was used for discrete covariates. During the incorporation of the covariates into the model, they were centered with the sample median. The selection of covariates was performed with the assistance of the COSSAC algorithm [29], taking into consideration a decrease in the objective function by −3.84 (p<0.05) for forward inclusion, and a change of +6.64 (p<0.01) for the backward deletion. For the final pharmacokinetic model, the relationship between variables and pharmacokinetic parameters was determined using criteria such as: (i) improvement in the model’s goodness-of-fit and (ii) biological plausibility of the relationship.

The predictive performance of the model was evaluated with a prediction corrected Visual Predictive Check (pcVPC) [30]. It was performed simulating the study design 1000 times for each simulation, and the median and the prediction intervals of 80% (10th, 50th, and 90th percentiles) were obtained. The median and 95% confidence intervals (2.5th, 50th, 97.5th) were determined for each prediction interval. We did simulations during a dosage interval of 0 to 8 h, with 1000 points and 100 bins, while, for the observations, we obtained the 80% empirical intervals taking six bins with an equivalent size.

### 2.6. Pharmacodynamic Evaluation

The following targets were evaluated for the PK/PD index: $60\%fT_{>\mathrm{MIC}}$ (cefepime-free concentration is maintained above the MIC value for at least 60% of the dosing interval), and $100\%fT_{>\mathrm{MIC}}$ (free concentration of cefepime is maintained above the MIC value throughout the dosing interval). The concentration of free cefepime was estimated, assuming a protein binding of 20% [31]. MIC was evaluated from 0.001 to 1024 mg/L; for each condition and target, 2500 virtual individuals were simulated with the final model using the R package MlxR. The evaluation of the PK/PD index was performed numerically through the proportion of (simulated) points above each MIC, with an algorithm implemented in the RcppArmadillo package of R [32].

Pharmacokinetic profiles were evaluated with administration in intermittent infusions (II), prolonged infusions (PI), or continuous infusions (CI). The administration regimens for II were 1 g every 6 h (q6h), 2 g every 12 h (q12h), 2 g every 8 h (q8h), 2 g q6h, and 4 g q12h; by default, the infusion time was assumed as 30 min. The PI administration regimens were 2 g q12h, 2 g q8h (2 h infusion), 2 g q8h, and 4 g q12h; by default, the infusion time was 4 h. The CI administration regimens were 4 g every day (q24h), 6 g q24h, and 8 g q24h in administration for 24 h, after a loading dose of 2 g infused for 30 min. A loading dose was applied because the steady state is reached only after 5 h from the start of the infusion, and there would be insufficient protection in this time period.

To visualize the impact of serum creatinine level on PTA, a dosing regime of 2 g q8h as an infusion of 30 min was evaluated for the two PK/PD targets. We evaluated levels from 0.2 to 1.0 mg/dL, within the range of $S_{\mathrm{CR}}$ values determined in patients in this study. Likewise, a scenario where the $S_{\mathrm{CR}}$ level is unknown was evaluated, and this was obtained by simulating a normal distribution of $S_{\mathrm{CR}}$ values with a mean and variance of 0.550 and 0.179 mg/dL, respectively.

A dosing regimen with probability of target attainment (PTA) greater or equal than 0.9 was considered optimal. The cumulative fraction of response (CFR) was determined with dosing regimens at steady state for *Escherichia coli*, *Klebsiella pneumoniae*, and *Pseudomonas aeruginosa* based on the MIC distribution of EUCAST (the European Committee for Antimicrobial Susceptibility and Testing) database (available at: https://mic.eucast.org/Eucast2/ accessed on 30 June 2020). The CFR was calculated for: (i) empirical therapy, taking the entire MIC distribution and (ii) directed therapy, taking the sensitivity MIC range below the cut-off points (4 mg/L for Enterobacteriaceae, and 8 mg/L for *P. aeruginosa*, according to EUCAST [33]).

### 2.7. Statistical Analysis

The correlation between dose and response (inhibition halos) was evaluated by means of linear least-squares regression adjustment. An exploratory data analysis of the covariate data was carried out employing the R^®^ programming language, version 3.6.0 (R Foundation for Statistical Computing, Vienna, Austria). The package “tidyverse” (Version 1.3.0.) in R^®^ was used to manipulate the data, and create the plots. Subject covariates were summarized by median and interquartile range for continuous data, and by number and percentage of subjects for categorical data. The correlation between the covariates and the individual pharmacokinetic parameters was evaluated using linear regressions and generalized additive models using the package mgcv in R^®^.

## 3. Results

The study cohort of pharmacokinetic parameters consisted of 15 adult patients, and six serum samples were collected from each of the patients in the period between June and December of 2015. The patients of the study were 10 men at the age range between 21 and 60 years (median 39), their weight was between 59 and 90 kg (median 65), seven patients had acute lymphoblastic leukemia, four myeloid leukemia (three acute and one chronic), three non-Hodgkin lymphoma, and one patient had multiple myeloma.

Albumin values were between 2.00 and 4.30 g/dL (median 3.35). The most frequently used chemotherapy regimens were GRAALL [34] in four patients, HyperCVAD [35] in two patients and a different chemotherapy regimen in nine patients, four patients (28.6%) received antimicrobial therapy in the month prior to admission due to the presentation of the CIFN event, three of them received a β-lactam, and none of them received cefepime during this time interval. Table 1 shows the general characteristics of the patients included.

### 3.1. Pharmacokinetic Results

A two-compartment linear elimination model adequately described the concentration-time (TSFD, *time since first dosage*) data. The residual variability was well described with an additive error model. The parameter Q was not estimated by the SAEM algorithm and fixed at a value of 23.4L/h (as the global minimum of the likelihood function) to improve the model’s stability (condition number).

The relationship between the continuous covariate serum creatinine ($S_{\mathrm{CR}}$) and CL was the only that improved the goodness of fit of the model, while presenting biological plausibility. An improvement of the BICc value was obtained with an exponential relationship model of CL-$S_{\mathrm{CR}}$ with (BICc = 543.9) vs. the base model (BICc = 545.5). The inclusion of this covariate decreased $\omega_{\mathrm{CL}}^{2}$ from 26.0% to 22.7%. The estimated values for the final model, as well as their confidence intervals, are reported in Table 2. The pcVPC (see Figure 1) shows a good predictive performance by the model on the observations, and the TSFD data were changed by TAD (*time after dosage*) to improve visualization.

### 3.2. Pharmacodynamic Results

The differences in PTA evaluated at “first 24 h” of treatment and “steady state” were predicted to be negligible. The MIC-based PTA for both targets at the steady state is observed in Figure 2; these results were obtained for virtual individuals with $S_{\mathrm{CR}}$ of 0.54mg/dL (median value), 2 PK/PD targets, and 12 dose regimens with FEP. CI regimens resulted in higher PTA than EI or II regimens in the various groups of daily doses. The cut-off MIC value (PTA > 0.9) for a dose of 2 g q8h in an infusion of 30 min was 0.5mg/L (for $100\%fT_{>\mathrm{MIC}}$) and 2mg/L (for $60\%fT_{>\mathrm{MIC}}$). When the infusion time increased to 2 h, the cut-off MIC value increased to 1.0mg/L (for $100\%fT_{>\mathrm{MIC}}$) and 4mg/L (for $60\%fT_{>\mathrm{MIC}}$). When the infusion time increased to 4 h, the cut-off MIC value increased to 2.0mg/L (for $100\%fT_{>\mathrm{MIC}}$) and 8mg/L (for $60\%fT_{>\mathrm{MIC}}$). With a continuous infusion of 6 g in 24 h, the cut-off value increases to 8mg/L (for both targets), and this is the highest cut-off MIC value obtained, even increasing the daily dose.

The PTA as a function of MIC (at steady state) for several values of $S_{\mathrm{CR}}$ is depicted in Figure 3. For a pathogen with a MIC of 0.5 mg/L, the dose regimen of 2 g q8h as an infusion of 30 min, resulted in a PTA below 0.9 for patients with $S_{\mathrm{CR}}$ below 0.2 mg/dL (for $100\%fT_{>\mathrm{MIC}}$). A higher serum creatinine level was related to a higher PTA; a patient with $S_{\mathrm{CR}}$ of 1.0 mg/dL had a cut-off MIC value of 2 mg/L ($100\%fT_{>\mathrm{MIC}}$). For a pathogen with a MIC of 2 mg/L, the regimen of 2 g q8h as an infusion of 30 min resulted in a PTA below 0.9 for patients with $S_{\mathrm{CR}}$ below 0.2 mg/dL (for $60\%fT_{>\mathrm{MIC}}$). The cut-off MIC value was 8 mg/L for a CI dose regimen of 6 g q24h, regardless of the $S_{\mathrm{CR}}$ value of the simulated individuals or PK/PD target (results not shown).

Table 3 shows the CFR results for virtual individuals with unknown $S_{\mathrm{CR}}$ (simulated from a probability distribution) for three microorganisms (*E. coli*, *K. pneumonia*, and *P. aeruginosa*) selected for their high incidence in patients with CIFN. The table shows a comparison of results in empirical and directed therapy with various administration regimens.

## 4. Discussion

The methodology used in this study to measure the serum concentrations of the antibiotic is based on the diffusion of the antibiotic in agar plates and the determination of diameters in inhibition halos [23]. This method was implemented because it is cheap, fast, and reproducible for our medium compared to chromatographic techniques such as HPLC [23,24]. In the pharmacokinetic analysis of these data, the assumption that the patients were in a stationary state was followed, and an infusion time of 30 min used for this group of patients was also considered.

A two-compartment model is chosen to explain the behavior of cefepime in this group of patients with cancer and CIFN. The PK model turned out to be similar to that described by Rhodes N. et al. [4], to that described by Sampol E. in patients with burns [8], and that described by Tam et al., in patients with various degrees of renal failure [36], different from that observed by the group of Roos et al., who described a three-compartment model in ICU patients [37], or the single-compartment model described by Whited et al., in patients with CIFN [5]. The heterogeneity found in the description of the pharmacokinetic model of cefepime makes it challenging to compare PK parameters between studies, and therefore between different populations.

Significant differences were found in PK parameters reported in the study of Rhodes et al. [4]. In this study, a population model of cefepime was obtained for CIFN patients with the following estimated parameters: CL6.33L/h, Q6.87L/h, $V_{1}$14.8L, and $V_{2}$10.9L. The estimated values were much lower than those reported in Table 2. Moreover, the study of Sime et al. [6] found the following parameters by a non-compartmental analysis of pharmacokinetics in CIFN patients with $V_{\mathrm{SS}}=33.4\;L$ and CL=8.6L/h, the value of $V_{1}$ estimated in this study was greater than the value of $V_{\mathrm{SS}}$ reported.

Patients with CIFN, compared to healthy volunteers, have a higher $V_{1}$ (26.43 L vs. 18.4 L), explained by a lower $C_{\max}^{\mathrm{SS}}$ (61.53 vs. 137 μg/mL), differences in CL are also observed for individuals with CIFN 12.88 L/h, concerning healthy volunteers 8.55 L/h [31]. These data could explain the clinical results identified in meta-analyses that have evaluated the impact of the use of cefepime in different patient groups [38], or in patients with neutropenia [6]. Although the increased mortality of cefepime in this setting remains controversial [6], it is evident that the meta-analyses start from diverse populations, and even groups of patients with different pharmacokinetic behavior [38,39].

Compared to other studies with neutropenic patients, there is a $V_{1}$ higher in this study of 26.43±4.01L than that reported by Whited et al. [5] of 20.9±1.3L. The elimination constant $k_{10}$ it was similar in this study 0.49±0.07$h^{-1}$ as reported by Whited et al. [5] with $k_{10}$ from 0.39±0.03$h^{-1}$. The results of a study carried out at Queen Elizabeth Hospital in Australia by Sime et al. [6] in a population with CIFN they were similar to the present study with CL of 8.6 vs. 12.88 L/h, for the reference and present study, respectively.

The Clinical and Laboratory Standards Institute (CLSI) defines several susceptibility cut-off MIC values for Enterobacteriaceae, with a susceptibility category with MIC < 2μg/mL for FEP, and a dose-dependent susceptibility category between 4 and 8μg/mL [40]. The breakpoint tables for interpretation of MIC values from EUCAST [33] defines other cut-off MIC values for cefepime in infections by Enterobacteriaceae susceptible (≤1mg/L) and resistant (≥4mg/L). In our study, for patients with an average $S_{\mathrm{CR}}$ of 0.54mg/dL, a standard dosing regimen of 2 g q8h in an infusion of 30 min reaches a PTA of 0.347 and 0.887 for the PK/PD indicators ($100\%fT_{>\mathrm{MIC}}$ and $60\%fT_{>\mathrm{MIC}}$) with a MIC of 4μg/mL, and, for this reason, such therapy would be inefficient. Increasing the infusion time to 2 h allows an acceptable PTA of 0.961 for the target $100\%fT_{>\mathrm{MIC}}$, while, for target $60\%fT_{>\mathrm{MIC}}$, a regimen of CI (6 g q24h) is required.

Various studies have found that renal function is a factor affecting FEP pharmacokinetics [36,41,42]. In this study, we observed that an increase in the serum creatinine ($S_{\mathrm{CR}}$) level is related to an increase in FEP clearance, and this with a decrease in the plasma FEP concentrations. The population pharmacokinetic model described here would be useful in predicting the pharmacokinetics of FEP in patients, in institutions where β-lactam monitoring is not available. In our study, we did not find a statistical relationship between FEP clearance and eGFR, and this result could be due to a lack of direct measurement of urine creatinine clearance.

For CFR analysis, directed therapy reflects antibiotic’s use when the infection-causing microorganism has been identified, as well as its susceptibility. Virtual individuals treated with cefepime regimens with a daily dose of 6g achieve the PK/PD target of $100\%fT_{>\mathrm{MIC}}$ in empirical therapy against *E. coli*. In comparison, the PK/PD targets of $60\%fT_{>\mathrm{MIC}}$ and $100\%fT_{>\mathrm{MIC}}$ are not achieved in the empirical therapy against *K. pneumoniae*, and, if an infection with this bacterium is suspected, a continuous infusion (CI) regimen should be used, in conjunction with continuous evaluation of possible signs of clinical deterioration. In the case of directed therapy against *K. pneumoniae*, regimens of intermittent infusion (II) should be used.

If suspicion of *P. aeruginosa* infection exists, CI dose regimens could be used to improve exposure. In the case of directed therapy against *P. aeruginosa*, II dose regimens could be used, but, if the patient has severe to absolute neutropenia, continuous infusion regimens could be used to maximize exposure.

In conclusion, this study shows a two-compartment model that adequately describes cefepime concentration profiles. More research is needed in subgroups of interest, among cancer patients and CIFN, for example, high- and low-risk patients, to find better ways to dose according to the type of population. The simulations applied in the study suggest the use of FEP as CI in empirical therapy when infection by bacteria with high resistance patterns, such as *K. pneumoniae* or *P. aeruginosa*, is suspected. Some other antibiotics (i.e., carbapenems) should be used when it is considered a PK/PD target of $100\%fT_{>\mathrm{MIC}}$ (which may be necessary in patients with severe neutropenia), and an infection by *K. pneumoniae* or *P. aeruginosa* is suspected.

## Figures and Tables

**Figure 1 antibiotics-10-00504-f001:**
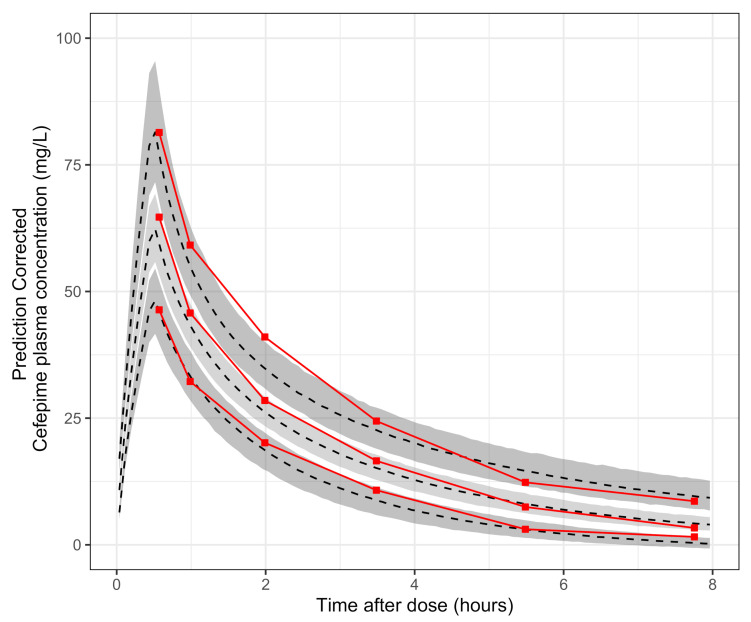
Prediction corrected Visual Predictive Check (pcVPC); red solid line, 10th, 50th, 90th percentiles of the observations; red solid dots, central location of bins used to calculate percentiles; gray dashed line, 10th, 50th, 90th prediction intervals of simulations; shaded area 95% confidence interval from prediction intervals obtained from simulations.

**Figure 2 antibiotics-10-00504-f002:**
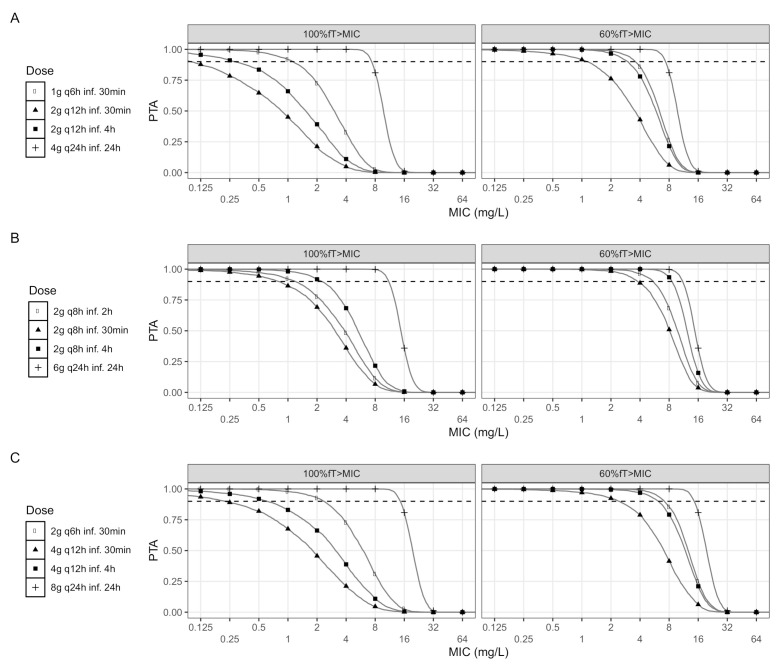
Probability of target attainment (PTA) for each of the 12 regimens (and the 2 PK/PD targets) using the final model and the population median value of $S_{\mathrm{CR}}$ (0.54mg/dL). The graphs indicate the total administration of 4 g (**A**), 6 g (**B**), or 8 g (**C**). Dashed lines, 90% of the simulated patients reached the specified target; $fT_{>\mathrm{MIC}}$, the percentage of the dosing interval that the free drug concentration is maintained above the MIC.

**Figure 3 antibiotics-10-00504-f003:**
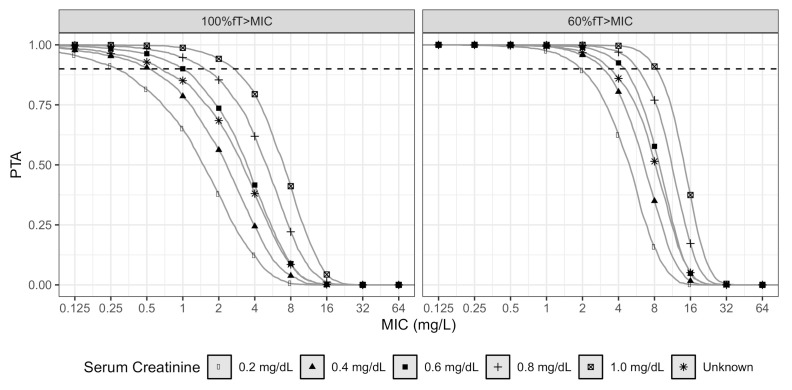
Probability of target attainment (PTA) versus MIC for different estimates of creatinine level, each derived from 2500 simulations of data (PTA was evaluated at steady state). The gray lines demonstrate PTA in various scenarios of creatinine level. The dotted black lines represent a PTA of 0.90.

**Table 1 antibiotics-10-00504-t001:** Demographic, clinical, and laboratory characteristics of the patients.

Parameter	Value
Age (years) [median (IQR)]	39.0 (30.0, 49.5)
Sex [n (%)]	
Female	5 (33.3)
Male	10 (66.7)
Body weight (kg) [median (IQR)]	65 (60.5, 72.0)
Height (cm) [median (IQR)]	170 (160, 175)
BMI (kg/m${}^{2}$) [median (IQR)]	23.9 (23.1, 25.8)
Serum creatinine (mg/dL) [median (IQR)]	0.54 (0.48, 0.57)
eGFR ${}^{a}$ ($\mathrm{mL}/\min/1.73\;m^{2}$) [median (IQR)]	129 (118.9, 138.8)
Albumin (g/dL) [median (IQR)]	3.4 (3.21, 3.66)
Total protein (g/dL) [median (IQR)]	5.9 (5.46, 6.1)
Antibiotic given in the previous month [n (%)]	
Yes	4 (28.6)
No	11 (71.4)
Concomitant antimicrobial prophylaxis [n (%)]	
Trimethoprim-Sulfametoxazole	3 (20.0)
Acyclovir	6 (40.0)
Fluconazole	2 (13.3)
Nystatin	2 (13.3)
Ivermectin	1 (6.7)
Metronidazole	1 (6.7)
Voriconazole	1 (6.7)
None	4 (26.7)
Malignancy [n (%)]	
Lymphoma	3 (20.0)
Lymphoid leukemia	7 (46.7)
Myeloid leukemia	4 (26.7)
Multiple myeloma	1 (6.7)
ANC (/μL) [median (IQR)]	30 (20, 95)
Therapy cycle [n (%)]	
1	11 (73.3)
2	3 (20.0)
>3	1 (6.7)
FN associated chemotherapy [n (%)]	
GRAALL ${}^{a}$	4 (26.7)
HyperCVAD ${}^{a}$	2 (13.3)
Another chemotherapy	9 (60.0)

^*a*^ see definition in Section 2.1, Section 2.5 and Section 3.

**Table 2 antibiotics-10-00504-t002:** Population pharmacokinetic parameters of cefepime in patients with CIFN.

Parameter	Estimated Value	RSE (%)	Bootstrap Median (95% CI) ${}^{a}$
**Structural Model**			
$\mathrm{TVCL}=\theta_{0}\times \exp\left(\theta_{1}\times S_{\mathrm{CR}}/0.47\right)$			
$\theta_{0}\;\left(L/h\right)$	20.6	19.67	20.72 (11.96, 33.54)
$\theta_{1}$	−0.415	44.08	−0.42 (−0.80, 0.17)
$V_{1}\;\left(L\right)$	23.8	9.47	24.04 (19.74, 28.99)
Q(L/h)	23.4	-	-
$V_{2}\;\left(L\right)$	13.3	28.38	12.79 (7.13, 22.59)
**Interindividual variability**			
$\omega_{\mathrm{CL}}^{2}\;\left(\%\right)$	22.70	19.15	21.2 (14.07, 26.5)
$\omega_{V_{1}}^{2}\;\left(\%\right)$	30.60	25.66	28.6 (0.000, 46.2)
$\omega_{Q}^{2}\;\left(\%\right)$	120.30	39.14	97.9 (0.008, 291.3)
$\omega_{V_{2}}^{2}\;\left(\%\right)$	93.90	30.44	86.9 (0.000, 160.8)
**Residual Error**			
Additive (mg/L)	1.86	11.11	1.89 (1.40, 2.40)

^*a*^ percentile confidence intervals and median were estimated with 1000 resampled datasets.

**Table 3 antibiotics-10-00504-t003:** Comparison of CFR for cefepime in empirical and directed therapy for three microorganisms. The gray color in a cell indicates a CFR ≥0.85.

Dose	Therapy	Microorganism ${}^{a}$		
		ECO	KPN	PSA
Target: $100\%fT_{>\mathrm{MIC}}$				
2 g q8h tinf 30 min	Empirical	0.899	0.610	0.396
	Directed	0.984	0.957	0.626
2 g q8h tinf 2 h	Empirical	0.910	0.631	0.463
	Directed	0.993	0.974	0.719
2 g q8h tinf 4 h	Empirical	0.920	0.657	0.565
	Directed	0.998	0.991	0.849
6 g q24h tinf 24 h	Empirical	0.942	0.749	0.820
	Directed	1.000	1.000	1.000
Target: $60\%fT_{>\mathrm{MIC}}$				
2 g q8h tinf 30 min	Empirical	0.928	0.686	0.663
	Directed	0.999	0.997	0.939
2 g q8h tinf 2 h	Empirical	0.933	0.706	0.724
	Directed	1.000	1.000	0.981
2 g q8h tinf 4 h	Empirical	0.939	0.730	0.785
	Directed	1.000	1.000	1.000
6 g q24h tinf 24 h	Empirical	0.942	0.749	0.820
	Directed	1.000	1.000	1.000

^*a*^ ECO: *Escherichia coli*, KPN: *Klebsiella pneumoniae*, PSA: *Pseudomonas aeruginosa*.

## Data Availability

The data presented in this study are available on request from the corresponding author.

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
