# Peer review of "Pharmacokinetics and Pharmacodynamics of Cefepime in Adults with Hematological Malignancies and Febrile Neutropenia after Chemotherapy"

_antibiotics, 2021, doi:10.3390/antibiotics10050504_

Round 1

Reviewer 1 Report

Cortés et al reported a two-compartment model that describe the pharmacokinetics and pharmacodynamics of the 4th generation antibiotic Cefepime in patients with hematological malignancies and febrile neutropenia after chemotherapy treatment. Additionally, the authors showed that the continuous infusions of Cefepime 6g q24h could have a good achievement of PK/PD target for MIC levels below the resistance cut-off point of Enterobacteriaceae. This study is very interesting and add a significant value to the scientific pharmacological field and specially to antibiotic field. However, there are some points need to be address before publishing this study:

  • The abstract is poorly written and contains several mistakes. Please, re-write in clarified way that clearly shows study overview, what have been done, and conclusion.
  • The introduction is very short and does not cover the topic, missing the citation of relevant studies.
  • The SD or SE must be presented for all experiments (fig 2 and 3)
  • The main argue here is the sample size (15 patients), how authors calculated the sample size?
  • What could be the PD/PK of cefepime in other type of cancers?
  • The conclusion part is very short, please modify it.
  • How this PD and PK study could be applied for other types of antibiotics?
  • please include the ethical agreement from your institute.

Reviewer 2 Report

From a clinical/epidemiological point of view, I would like to pose some questions to the Authors

  • everywhere: please, report all continuous covariate result as median/IQR and not mean/sd; redo inferential analyses according to a non-parametric approach
  • line 60: since you enrolled naive pts, you can't state "maintenance or relapse chemotherapy",  it's counterfactual!
  • line 62: are fever and neutropenia definitions compliant with those from CTCAE? please, provide details
  • lines 136-7: " administration in intermittent infusions (II), prolonged infusions (PI), or continuous infusions (CI)" how the Colombian drug manufacturer suggest to administer cefepime, since you tested 3 different approach?
  • line 163 exploratory data analysis, please provide more details (descriptive and inferential tests applied / R package used)
  • line 171:  end of study December 2015: what's the matter with this long delay?
  • table 1: what about potential pk/pd interactions with concomitant administered drugs? Have you considered them?

Round 2

Reviewer 1 Report

Thanks to the authors for addressing all the questions raised. I have only two minor points:

1- the introduction: still authors did not cite the relevant studies. for example,

''Knowledge of the PK parameters ofβ-lactam antibiotics in patients with32CIFN has expanded in recent years and the variability that such parameters present in33this special group have been described concerning other groups of critically ill patients,34such as those who receive cefepime in burn units or the ICU''

''Cefepime (FEP) is a cephalosporin with a broad spectrum of activity and good36penetration into various tissues''

''For patients with CIFN, it is not entirely clear what is the PDI target associated40with the clinical cure, microbiological eradication, or survival.''

and so on...

2- the MS still contains many languages mistakes. Please, carefully edit the manuscript.

Author Response

Report 1

Thanks to the authors for addressing all the questions raised. I have only two minor points:

Point 1: the introduction: still authors did not cite the relevant studies. for example,

''Knowledge of the PK parameters ofβ-lactam antibiotics in patients with32CIFN has expanded in recent years and the variability that such parameters present in33this special group have been described concerning other groups of critically ill patients,34such as those who receive cefepime in burn units or the ICU''

''Cefepime (FEP) is a cephalosporin with a broad spectrum of activity and good36penetration into various tissues''

''For patients with CIFN, it is not entirely clear what is the PDI target associated40with the clinical cure, microbiological eradication, or survival.''

and so on...

Response 1: We thank you for the suggestion on this subject, the following references were added in the introduction.

  • "Knowledge of the PK parameters of β-lactam antibiotics in patients with CIFN has expanded in recent years [1–3] and the variability that such parameters present in this special group have been described concerning other groups of critically ill patients, such as those who receive cefepime in burn units or the ICU [3]." [Lines 32-35]
  • "Cefepime (FEP) is a cephalosporin with a broad spectrum of activity [4] and good penetration into various tissues [5]. " [Lines 36-37]
  • "For patients with CIFN, it is not entirely clear what is the PDI target associated with the clinical cure, microbiological eradication, or survival [6]." [Lines 40-41]

2- the MS still contains many languages mistakes. Please, carefully edit the manuscript.

Response 2: We thank you for the suggestion. Below we show, the changes in the wording after the review by the language editing service:

Line

Old

New

30

… β-lactams and vancomycin often required for the management of this special group..

…β-lactams and vancomycin often required for the management of this particular group…

40

For patients with CIFN, it is not entirely clear what is the PDI target associated with…

For patients with CIFN, it is not entirely clear the PDI target associated with…

43

…associated with a lower risk of negative outcomes related to bacteriological eradication…

…associated with a lower risk of adverse outcomes related to bacteriological eradication…

72-73

…(v) major comorbidities in addition to its underlying neoplasm...

…(v) significant comorbidities in addition to its underlying neoplasm…

78-79

dose of 2g every 8 hours, which was administered in a 30-minute infusion

…dose of 2g every 8 hours, administered in a 30-minute infusion…

80

The infusions were made by means of an infusion pump.

The infusions were made using an infusion pump.

120-122

The covariates of the final model were determined by their improvement in the goodness of fit of the model, as well as the biological plausibility of the relationship between the covariate and the pharmacokinetic parameter.

For the final pharmacokinetic model, the relationship between variables and pharmacokinetic parameters was determined using criteria such as: (i) improvement in the model’s goodness-of-fit and (ii) biological plausibility of the relationship.

126

This was performed simulating the study design

It was performed simulating the study design

152

To visualize the impact of serum creatinine level on PTA, a dosing regimens of 2g

To visualize the impact of serum creatinine level on PTA, a dosing regimen of 2g

154

..We evaluated levels from 0.2 to 1.0 mg/dL, which are within the range of SCR values determined in patients in this study…

..We evaluated levels from 0.2 to 1.0 mg/dL, within the range of SCR values determined in patients in this study…

196

…(as the global minimum of the likelihood function) to improve the stability (condition number) of the model.

…(as the global minimum of the likelihood function) to improve the model's stability (condition number).

198

…was the only that improved the goodness of fit of the model, and also had biological plausibility

…was the only that improved the goodness of fit of the model, while presenting biological plausibility.

References

  1. Rhodes NJ, Grove ME, Kiel PJ, O’Donnell JN, Whited LK, Rose DT, et al. Population pharmacokinetics of cefepime in febrile neutropenia: implications for dose-dependent susceptibility and contemporary dosing regimens. International Journal of Antimicrobial Agents. Elsevier B.V.; 2017;50:482–6.
  2. Whited L, Grove M, Rose D, Rhodes NJ, Scheetz MH, O’Donnell JN, et al. Pharmacokinetics of Cefepime in Patients with Cancer and Febrile Neutropenia in the Setting of Hematologic Malignancies or Hematopoeitic Cell Transplantation. Pharmacotherapy: The Journal of Human Pharmacology and Drug Therapy. 2016;36:1003–10.
  3. Sime FB, Roberts MS, Tiong IS, Gardner JH, Lehman S, Peake SL, et al. Adequacy of High-Dose Cefepime Regimen in Febrile Neutropenic Patients with Hematological Malignancies. Antimicrobial Agents and Chemotherapy. 2015;59:5463–9.
  4. Patel HB, Lusk KA, Cota JM. The Role of Cefepime in the Treatment of Extended-Spectrum Beta-Lactamase Infections. Journal of Pharmacy Practice. 2019;32:458–63.
  5. Sampol E, Jacquet A, Viggiano M, Bernini V, Manelli J-C, Lacarelle B, et al. Plasma, urine and skin pharmacokinetics of cefepime in burns patients. Journal of Antimicrobial Chemotherapy. 2000;46:315–7.
  6. Burgess S V., Mabasa VH, Chow I, Ensom MHH. Evaluating Outcomes of Alternative Dosing Strategies for Cefepime: A Qualitative Systematic Review. Annals of Pharmacotherapy. 2015;49:311–22.